

# The Effect of Community Resilience and Disaster Risk Management Cycle Stages on Morbi-Mortality Following Floods: An Empirical Assessment

Raquel Guimaraes[1], Reinhard Mechler[1], Stefan Velev[1], Dipesh Chapagain[1,2]

[1]International Institute for Applied Systems Analysis, Laxenburg, 2361, Austria.
[2]United Nations University Institute for Environment and Human Security (UNU-EHS), Bonn, 53113, Germany.

*Correspondence to*: Raquel Guimaraes (guimaraes@iiasa.ac.at)

**Abstract.** Practice and policy have emphasised the need for building resilience to climate-related events in a
further warming world. Scholarship has studied resilience largely in terms of process, latent capacity informing vulnerability, or outcome of risk management interventions, with little work integrating these perspectives. Implementation science work by the Climate Resilience Alliance has developed the Flood Resilience Measurement for Communities (FRMC) process and tool to measure resilience as outcome (post-flood mortality and morbidity reduction) and as capacity (pre- and post-intervention levels). This article builds on FRMC
analytics to investigate the effect of resilience capacity, represented by five capitals, and five stages of the Disaster Risk Management Cycle (DRM), on injury and mortality outcomes across 66 flood-affected communities in seven Global South countries. Using a quasi-experimental design with regression adjustment, we analyse the relationship between resilience levels, DRM stages, and health outcomes. Results show that social and human capital help reduce injuries after floods, and preparedness lowers both deaths and injuries.
Some results were unexpected, such as the positive association between natural capital and delayed deaths, where limited gains in natural capital may not yield meaningful protection in communities with degraded ecosystems.

## 1 Introduction

Floods are the most frequent disaster triggered by environmental extremes and account for the highest disaster-
related death rate (Yari et al., 2020). Also, floods cause severe health impacts worldwide, particularly affecting lower-income, densely populated regions (Escobar Carías et al., 2022; Lynch et al., 2025). Research highlights that mortality and morbidity during and after floods are shaped by a variety of individual and community risk factors, including hazard event type, intensity and duration (Birkmann et al., 2022) and factors associated with exposure and vulnerability including age (Petrucci, 2022; Yang et al., 2023), gender (Jerin et al., 2024;
Mucherera and Mavhura, 2020), urban-rural location (Petrucci, 2022), and various other drivers, such as access to finance post-disaster, hazard awareness, quality of early warning and disaster response – all of which significantly determine risk and actual impacts of affected populations in disaster events as well as inform interventions to build resilience (Chapagain et al., 2024b).

Although the impact of demographic factors – such as age, gender, and rural or urban residence – on flood-
related mortality and morbidity is well-documented, the role of community-level resilience and stages of DRM cycle, when accounting for demographic factors and flood exposure/hazard, is less clear. One study has



explored the role of five capitals (social, human, financial, physical, natural) as drivers of vulnerability on total injuries and fatalities, controlling for flood exposure, but with no control for demographics (Chapagain et al., 2024b). Another study analysed the obstacles encountered at different stages of the disaster risk management cycle concerning rural flooding in Pakistan. Employing a qualitative approach with focus groups and key informant interviews in the Khyber Pakhtunkhwa province, the study identified challenges in risk reduction, preparedness, rescue and relief, and rehabilitation and recovery phases (Shah et al., 2023).

Understanding resilience has proceeded. Drawing on Alexander (2013), resilience has evolved from an outcome-oriented concept - "bouncing back" after disturbance - to one that also includes latent features such as inherent capacity and adaptive potential. Initially defined by its etymology and early scientific use in mechanics (resisting and absorbing force), resilience concepts have been advanced to reflect deeper systemic traits in ecology (absorbing shocks while maintaining function), psychology (individual adaptation to adversity), and the social sciences (community robustness and flexibility for transformation). In disaster risk reduction (and climate adaptation) research and implementation, dimensions have seen attention and resilience has been analyzed as observable outcomes after events and latent qualities before events - like adaptability, resistance, and transformative capacity - that enable systems to withstand and evolve through disruption (Alexander, 2013). Little work has integrated outcome and capacity, however.

Furthermore, understanding resilience and DRM stages effects and their interactions with demographic profiles and exposure are essential for developing effective policy interventions. Available evidence suggests that investing in building community resilience to floods reduces the negative impacts of these events on human health and well-being along a DRM cycle aims to avoid, lessen, or transfer the adverse effects of floods, contributing to better flood outcomes by guiding integrated and proactive management strategies (Hochrainer-Stigler et al., 2020, 2021; Keating et al., 2017a; Laurien et al., 2020).

This article uses the Flood Resilience Measurement for Communities (FRMC) tool to examine the role of capacities in reducing fatalities (immediate and delayed) and injury outcomes for 66 flood-affected communities across seven countries. Recent global analysis of the FRMC's large-scale application across nearly 400 communities further validates its use, highlighting consistent patterns in how different resilience dimensions relate to recovery outcomes (Keating et al., 2025).

The analysis controls for demographic factors and flood exposure/hazard. Communities are grouped into four resilience clusters and two DRM cycle groups to reflect the tendency of similar capital levels and DRM coping capacities (Hochrainer-Stigler et al., 2021; Keating et al., 2025). A quasi-experimental research design with regression adjustment is applied to evaluate the distinct influence of resilience and DRM cycle stages on morbidity and mortality outcomes after controlling for confounders.

Our findings emphasize the critical role of resilience and Disaster Risk Management (DRM) cycle stages in shaping health and mortality outcomes after a flood event. Social and Natural capital were assessed to be effective in reducing injuries. DRM models demonstrated stronger predictive power, with Preparedness significantly decreasing both fatalities and injuries. Meanwhile, Corrective Risk Reduction lowered injury rates. Nevertheless, we found some unexpected results. For instance, the positive association between natural capital and delayed mortality may reflect the vulnerability of communities with degraded ecosystems, where limited improvements fail to yield meaningful health benefits. Similarly, unexpected patterns in the DRM stages, such



as higher mortality associated with corrective risk reduction, may be explained by lagged effects or measurement timing relative to implementation efforts.

The paper is structured as follows: Section 2 outlines the conceptual frameworks, focusing on the FRMC five-capitals model and the DRM cycle stages. Section 3 reviews literature on factors influencing flood-related

mortality and morbidity, including demographic characteristics and flood exposure/hazard. Section 4 describes the FRMC data used for analysis. Section 5 details the methodology applied. Section 6 presents the study results, while Section 7 discusses the key findings and limitations of the study.

## 2 Conceptual Frameworks

This section outlines the conceptual frameworks used to analyse the impact variables, emphasizing their

theoretical components and relevant literature.

### 2.1 The FRMC capitals framework (5Cs)

The Flood Resilience Measurement for Communities (FRMC) framework is a tool designed to measure and analyse a community's resilience to flooding. The FRMC has been developed by the Flood Resilience Alliance (now Climate Resilience Alliance), in partnership of NGOs, humanitarian organizations, risk engineers, and

researchers. It is the world's most widely used and validated community-level disaster and climate resilience measurement approach and has been applied in over 400 communities across more than 40 countries across the globe. The FRMC builds on a decade of evidence and real world impact generated through a community-led approach for practically and holistically measuring resilience to multiple hazards (Campbell et al., 2019; Keating et al., 2017a, 2025). The FRMC is built upon a rich dataset compiling information from households,

key informants, focus groups and secondary sources.

A central component of the framework is the five capitals (5Cs), which are broadly derived from the Sustainable Livelihoods Framework (Keating et al., 2014). The 5Cs represent different types of assets and resources that contribute to a community's overall well-being and its capacity to cope with and recover from shocks, including floods. A summary of each capital is provided next (Campbell et al., 2019; Keating et al., 2017a).

Human Capital refers to the education, skills, health, and well-being of household members in a community that enhance their ability to prepare for and recover from a flood. Examples include flood preparedness knowledge, personal safety skills, and education levels. Social Capital encompasses the social relationships, networks, and shared norms that enable communities to support each other, such as formal community emergency services and community-led flood management efforts. Financial Capital includes the financial resources available to

households and communities, such as savings, income, access to credit, and government funding for infrastructure. Physical Capital consists of the built infrastructure essential for both daily life and disaster response, including roads, communication systems, and flood defences. Finally, Natural Capital encompasses the natural resources and ecosystems that provide flood protection and sustain livelihoods, such as wetlands, forests, and managed biodiversity.

Studies emphasize that these five capitals are interconnected and influence each other. For example, a community with strong social capital may be better able to mobilize financial resources or advocate for improved physical infrastructure. Similarly, investments in human capital, such as education and training, can





enhance a community's capacity to manage financial resources effectively or adopt flood-resilient building practices. The FRMC framework provides a comprehensive method for evaluating community flood resilience

by recognizing these interconnections (Campbell et al., 2019; Chapagain et al., 2024b; Hochrainer-Stigler et al., 2020, 2021; Keating et al., 2014).

The 5Cs can play a crucial role in influencing mortality and morbidity outcomes after a flood event. For example, it is expected that communities with strong physical infrastructure, such as well-maintained drainage systems and flood-resistant housing, will tend to experience fewer injuries and fatalities. Similarly, robust social

capital can support community-based early warning systems, aid vulnerable individuals, and ensure access to essential resources, potentially reducing health impacts. A study using FRMC data examined these effects and found that higher levels of natural, physical, and financial capital are linked to better resilience outcomes, resulting in reduced mortality and injury rates (Chapagain et al., 2024b). Specifically, communities with greater natural and physical capital reported fewer post-flood injuries and fatalities. Financial capital, particularly in

urban areas, contributed to supporting post-flood livelihoods and governance. Although social and human capital had a smaller effect, they still played an important role, especially in rural communities where social capital enhanced mutual aid and recovery efforts.

FRMC analysis can zoom into key phases of the DRM cycle, which is broken down into 5 stages (Keating et al., 2017a, b). Prospective Risk Reduction involves taking proactive steps to prevent new risks from arising.

Corrective Risk Reduction focuses on lowering risks for people and assets already at risk. Preparedness is about getting people and resources ready for possible events. Response encompasses the immediate measures implemented during and right after a disaster to reduce its effects. Recovery, on the other hand, includes both short- and long-term efforts aimed at supporting individuals and communities in managing the aftermath

This paper contributes to the existing literature by analysing the effects on mortality and morbidity outcomes

along resilience capitals and DRM cycle stage.

## 3 Review: measurement and indicators

### 3.1 Demographics and Morbi-mortality due to floods

#### 3.1.1 Gender and age

Research suggests that the relationship between flood-related morbidity and mortality and age, and gender is

complex as exposure and vulnerability factors often interact, with their impact on health and mortality frequently depending on the context.

Evidence on the gender-specific effects of floods on mortality is well-documented. Globally, men generally experience higher mortality rates during flood events (Jerin et al., 2024; Petrucci, 2022). An analysis of research conducted in Europe, the United States, and Australia found that 65% of the studies reported consistently higher

fatality rates among men. The study highlights that in the United States (1996–2014), male flood fatalities consistently outnumbered female ones across all scenarios. A similar pattern was observed in parts of Europe (1980–2018), where male fatalities were generally higher, except among the elderly. The review attributes this increased male vulnerability to greater exposure to flood hazards and the higher proportion of men who operate vehicles during such events (Petrucci, 2022).





Morbidity effects, on the other hand, tend to stress the vulnerability of women and of specific population age-groups. For instance, a study emphasizes the heightened health vulnerabilities of women during floods due to factors like polluted water and challenges in menstrual management (Jerin et al., 2024).

In contrast to gender influences, the effects of age on flood fatalities and injuries vary significantly across studies. Some research, for instance, emphasizes that older individuals are particularly prone to fatalities during

and in the aftermath of floods (Ban et al., 2023; Yang et al., 2023), while there is evidence indicating that younger individuals can face a higher risk of mortality, specifically non-accidental deaths, during flood events compared to older adults (Ban et al., 2023).

In sum, the literature indicates that the relationship between flood impacts and risk, age, and gender is multifaceted and requires further attention, also as some studies have suggested that women's access to human,

social, and financial resources can strengthen their ability to adapt to floods (Azad and Pritchard, 2023). This evidence is particularly relevant to our study, as we focus on the net effect of community resilience levels and DRM cycle stages on flood-related mortality and morbidity.

### 3.1.2 Urban-Rural residence

Studies show that geographic factors play a critical role in shaping flood-related mortality. Rural areas face

higher risks due to slower emergency response capabilities, lower population densities that limit immediate aid from bystanders, a lack of protective infrastructure such as raised bridges, and their frequent location in headwater basins where floods develop rapidly, allowing little time for warning or evacuation. In contrast, the study finds that the presence of more valuable assets, higher average incomes, and housing structures can contribute to greater resilience and a reduced concentration of risk in urban areas (Petrucci, 2022).

The intersection between demographic factors and urban/rural residence is also relevant. A study stresses that women in rural Bangladesh are more vulnerable to floods due to patriarchal norms that limit their access to resources and decision-making (Azad and Pritchard, 2023).

In summary, research suggests that while both urban and rural areas face flood risks, specific setting factors may lead to significantly differential impacts on mortality and morbidity. Households and communities in rural areas

often lack resources, infrastructure, and emergency response capabilities, making them more vulnerable to flood damage and associated health and mortality impacts. Urban areas, on the other hand, have been found to exhibit greater resilience due to better resources and infrastructure (Campbell et al., 2019), though specific vulnerabilities can exist within urban settings as well. For instance, dense urban development can lead to the substitution of natural flood risk reduction features within built infrastructure (Laurien et al., 2020, p.2). Further

research is needed to understand the complex interplay of factors shaping flood vulnerability across different geographical contexts and population groups.

### 3.2 The Impact of Flood Hazards and Exposure on Health and Mortality Outcomes using FRMC

Research indicates a strong relationship between the severity of a flood event – often measured by a flood's return period – and its health and mortality impacts. For instance, studies using FRMC data show that

communities hit by rare, catastrophic floods affecting large areas tend to report higher rates of injuries and property damage. Additionally, there is a solid connection between a flood's return period and a community's preparedness; communities frequently exposed to milder recurrent floods (e.g., 1–2-year return periods) may





develop adaptive behaviours and a higher preparedness level, which can reduce injuries and speed recovery. In contrast, infrequent but severe floods (e.g., 50 and 100+ year return periods) often overwhelm even resilient

communities, leading to more serious health and mortality consequences (Campbell et al., 2019; Chapagain et al., 2024b). Findings show also that extreme precipitation significantly increases mortality, with heavy rain days linked to a 45% rise in landslide mortality and a 33% increase in flood mortality (Chapagain et al., 2024c).

## 4 Data

The FRMC data set provides a rich and multifaceted view of flood resilience at community level, collected

through a standardized framework and tool. This approach uses a flexible, mixed-method strategy where trained practitioners, often NGO staff, gather information via household surveys for individual and household data, community focus group discussions for collective insights, key informant interviews with knowledgeable community leaders, focus group discussions to capture specific perspectives, and third-party data sources like census data, government reports, or academic studies (Campbell et al., 2019; Hochrainer-Stigler et al., 2020,

2021; Keating et al., 2014, 2017a, 2025; Laurien et al., 2020).

The FRMC framework includes three key phases of data collection: baseline, post-event, and endline. Each phase captures specific aspects of a community's resilience and DRM cycle levels, providing a comprehensive view of how communities prepare for, respond to, and recover from flood events.

*Baseline Data Collection (BL)* occurs before a flood event, establishing a snapshot of the community's initial

resilience and DRM capacity. This phase gathers data on the pre-existing sources of resilience across five capitals (human, social, physical, financial, and natural) and DRM cycle stages, offering a benchmark for tracking resilience changes over time and assessing intervention impacts. Data is collected through methods such as household surveys, community focus groups, key informant interviews, focus group discussions, and third-party sources like census or government reports. Collected data is graded from A (Best Practice) to D

(Significantly Below Good Standard, Potential for Significant Loss), with aggregated scores (A=100, B=67, C=33, D=0) providing an overall picture of the community's resilience capacity.

*Post-Event Data Collection (PE)* takes place after a flood impacts a participating community. Its purpose is to measure the actual effects of the flood on the community, as well as the resilience demonstrated through losses prevented and recovery speed. This data validates the effectiveness of resilience sources identified in the

baseline phase. The focus is on 29 outcome measures, including control variables (like flood severity), impact variables (such as injuries or property damage), and action variables (like early warning use). A full description of the variables in provided in the appendices. Similar data collection methods to the baseline phase are used, and assessors apply professional judgment to grade resilience outcomes (A-D) based on observed impacts, allowing a comparison with baseline data to evaluate how resilience factors contributed to reducing losses and

improving recovery.

*Endline Data Collection (EL)* is conducted one to two years after the baseline, regardless of flood events in the interim. This phase reassesses the community's resilience capacity and measures any changes in the resilience sources, providing insights into intervention effectiveness, if applicable. Data collection mirrors the baseline, with updates to the community's characteristics as needed.

Together, these three data collection phases offer a detailed, dynamic perspective on community flood resilience. By examining resilience before, during, and after flood events, the FRMC framework helps clarify



which factors strengthen a community's capacity to withstand floods, the success of implemented interventions, and how resilience evolves.

In this study, we focus on baseline (BL) and post-event (PE) data. BL data provides insights into pre-flood
resilience levels and DRM cycle stages indicators and the demographic profile of communities. PE data assesses three outcome variables: flood-related death counts, mortality due to illness within three months post-flood, and the number of injuries. Additionally, PE data is used to evaluate flood exposure (percentage of the community affected), and the flood return period.

In addition to deaths occurring during a flood event, examining post-flood mortality due to illness within the
following three months is also important – and can be effectively assessed using FRMC data. Evidence suggests that all-cause, cardiovascular, and respiratory mortality risks remain elevated for up to 60 days after flood exposure (Yang et al., 2023).

This study examines the resilience of 66 riverine flood-prone communities in seven developing countries (Table A1, appendices), each of which experienced a flooding event following the baseline assessment. The time span
for the start of post-event data collection ranges from 2019 to 2023. Although the data does not specify the exact date of the flood, post-event data was collected only from communities that experienced a flood after the baseline data collection.

## 5 Methods for data analysis

This section outlines the methods used in this study. We begin by describing the variables included in the
analysis. For the resilience measures (the 5Cs), a detailed list of variables for each construct is provided in Table A2 (appendices). For the Disaster Management Cycle stages, which in line with the literature is broken down into the stages of Prospective Risk Reduction, Corrective Risk Reduction, Preparedness, Response and Recovery, a description is available on Table A3 (appendices).

Outcome variables - mortality (immediate and delayed) and injuries - are presented in the questionnaire format
in Table A4 (appendices). Finally, a description of the control variables, including flood exposure, return period, and community-level demographic indicators, are listed in Table A5 (appendices).

In this analysis, variables gathered from different respondents (key informants, focus groups, and secondary sources) were averaged to produce a single response for each community. Household-level variables were also aggregated to the community level by averaging. One limitation is that demographic variables (age, gender, and
urban-rural composition) reflect only the respondent's information rather than all household members. Consequently, our approach provides an approximate demographic profile of each community.

### 5.1 Principal Component Analysis (PCA) of the FRMC Five Capitals and DRM cycle stages

To estimate the five capitals along the DRM cycle stages, we use a latent construct approach. Principal Component Analysis (PCA) is conducted to derive a single construct for each capital: social, physical, natural,
human, and financial, as well as along each of the phases of the DRM cycle (Prospective Risk Reduction, Corrective Risk Reduction, Crisis Preparedness, Response, Recovery). Components are weighted by estimated population (households multiplied by average household size) to account for varying community scales. Validation of these constructs starts with decomposing the correlation matrix into eigenvalues and eigenvectors,





and a screeplot helped determine the number of factors to retain. We then calculate the Cronbach's alpha
coefficients (Table A6 and A7, appendices) to ensure it meets the acceptable threshold of 0.7. All DRM stages
except Prospective Risk Reduction and Recovery met this criterion, which were close to 0.6. Histograms of the
constructs (resilience and DRM cycles) are included in the appendices (Figures A1 and A2), as well as their
corresponding screeplots (Figures A3 and A4).

### 5.2 Community clusters

### 5.1.1 Resilience levels

A study using the FRMC framework has emphasized the importance of clustering communities to better
understand how resilience changes over time (Chapagain et al., 2024a). Using hierarchical clustering methods,
the study identifies five distinct community clusters based on the five capitals scores. A summary of the
characteristics of the clusters is as follows. Table A8 (appendices) presents the distribution of the communities
across the clusters. Figure A5 in the appendices present the average score of the five capitals by cluster in the
baseline survey.

- Cluster 1: Features the lowest resilience across all capitals.
- Cluster 2: Exhibits marginally stronger performance in financial, human, and physical capital
  compared to natural and social capital.
- Cluster 3: Presents relatively high human, natural, and social capital scores, but lower financial and
  physical capitals.
- Cluster 4: Demonstrates generally higher average capital scores compared to Clusters 1-3, particularly
  in human, natural, and social capital.
- Cluster 5: Exhibits the highest financial and physical capitals, with an overall profile of high average
capital scores.

In line with this approach, we group the 66 communities which experienced a flood event into the five resilience
clusters. No community affected by flood was found in Cluster 5.

### 5.1.2 DRM cycle stages

We applied the same clustering methodology approach as Chapagain et al. (2024b) to the baseline data to
classify the communities based on their DRM cycle performance, maintaining consistency with our earlier
analysis approach. The dendogram revealed two distinct clusters according to DRM cycle stages. Figure A6 in
the appendices present the average score of the five stages by cluster in the baseline survey.

- Cluster 1: demonstrates strong capabilities across most dimensions of the DRM cycle. These
  communities exhibit above-average Preparedness. Their Protective Risk Reduction and their Recovery
capabilities are notably strong.
- Cluster 2: represent significant weaknesses across all measured DRM dimensions. These communities
  show poor Response and Recovery capabilities. Their Preparedness scores are substantially below
  average.



As made for the resilience clusters, we grouped the 66 communities which experienced a flood event into the two DRM cycle clusters. Table A8 (appendices) presents the distribution of the communities across the clusters.

### 5.3 Regression adjustment

To estimate the effect of the 5Cs and DRM cycle stages on morbidity and mortality outcomes, we use a quasi-experimental research design based on regression adjustment (Angrist and Pischke, 2009, 2014; Imbens and Rubin, 2015). This method is a robust approach for identifying causal effects in observational data by addressing the challenge of confounding. While the FRMC dataset includes a longitudinal component (baseline, post-event, and endline data for tracking communities), for the post-event (PE) data, we only have one time point available. This necessitates controlling for baseline levels of resilience, DRM cycle stages, demographics, and flood exposure/hazard.

Regression adjustment allows us to isolate the relationship of interest – the effect of the 5Cs and DRM cycle levels on morbidity and mortality indicators – by accounting for observable characteristics that could otherwise bias the estimated treatment effects. Properly specified regression models reduce systematic differences between units, thereby approximating *ceteris paribus* conditions and enhancing the validity of the causal inference.

The analysis focuses on three main dependent variables. The first and second are the average number of injuries and deaths reported for men, women, and children, based on combined data from key informants, focus groups, and secondary sources. The third is the average number of individuals who lost their lives due to illnesses within three months following the flood. To ensure robustness, post-estimation diagnostics were conducted, including tests for overall model significance (R-squared) and comparison of AIC and BIC values to evaluate model fit. Ordinary least squares (OLS) regression was used as the primary method, incorporating nested models to assess how additional predictors contributed to the model's explanatory power. We attempted to fit a zero-inflated Poisson model to address the high number of zeros in the mortality and morbidity counts. However, the model failed to converge, likely due to the small sample size of 66 communities. Finally, given the differing scales of predictors – such as the average return period (1 to 35) and the percentage of female household respondents (0 to 1) – all predictors were standardized to enable meaningful comparison and interpretation of their relative importance.

We use clustered standard errors to address the fact that observations within the same cluster might be like each other in terms of both resilience and DRM cycle levels. As we have a small number of clusters – 5 for resilience and 2 for the DRM cycle – we employ the wild bootstrap approach (Cameron et al., 2008). The wild bootstrap is primarily used to obtain more reliable inference – such as p-values and confidence intervals – by addressing issues like heteroskedasticity or a small number of clusters.

### 6 Results

Our analysis revealed distinct patterns in the relationships between the 5Cs, DRM cycle stages, and the morbidity and mortality outcomes, controlling for demographic characteristics and flood exposure/hazard. But let's first begin with a description of the data.

The distribution of the dependent variables – average injuries due to floods, average deaths, and average lives lost to illnesses within three months – is shown in histograms in Figures 1 and 2. Notably, all three variables





exhibit a high number of zeros, and the death counts are characterized by a low number of cases (maximum of 11 for immediate mortality and 23 for delayed mortality).

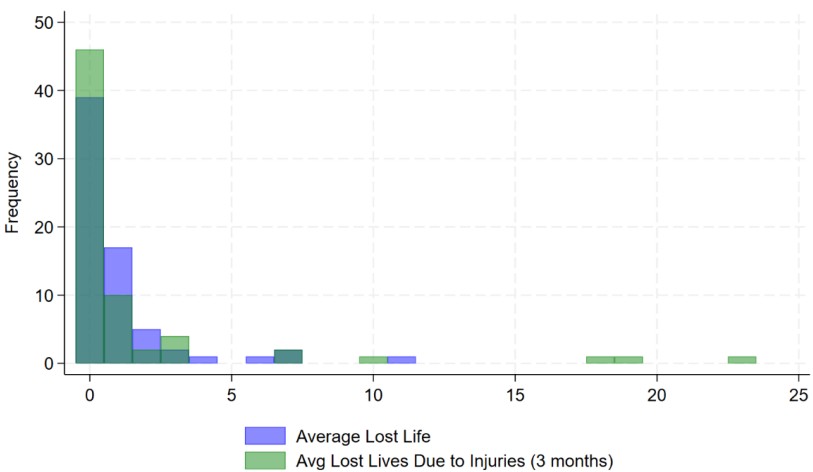

**340** **Figure 1: Average Number of Deaths (Immediate and Delayed) Reported in Flood-Affected Communities.**

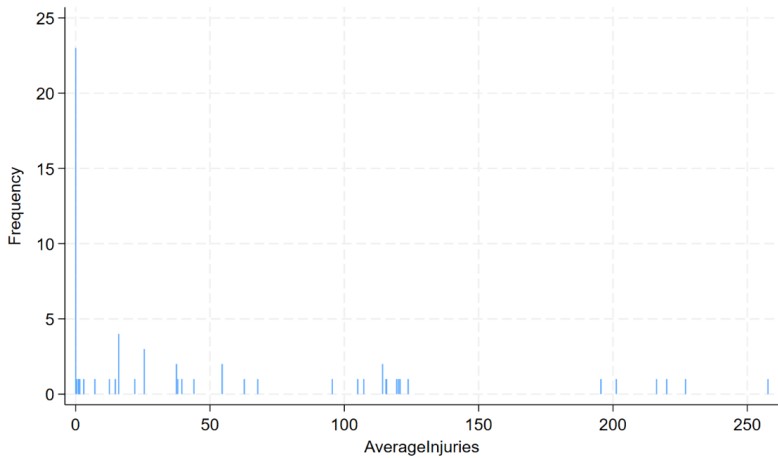

**Figure 2: Average Injuries Reported in Flood-Affected Communities.**

To facilitate interpretation, we use variable labels for statistics and regression results. We provide in Table 1 a
**345** description of the variables and in Table A9 (appendices) the descriptive statistics.

Table 1: Description of Variable Labels

| Variable | Description |
| --- | --- |
| AverageAge15to25 | The average proportion of the population aged 15 to 25 years in the community. |





| AverageAge26to50 | The average proportion of the population aged 26 to 50 years in the community. |
|---|---|
| AverageAge50plus | The average proportion of the population aged 50 years and above in the community. |
| AveragePercFemaleResp | The average percentage of female respondents in the community. |
| AverageRural | The average proportion of the community classified as rural. |
| AverageReturnPeriod | The average return period of significant flooding events in years for the community. |
| AveragePercComAffect | The average percentage of the community affected by the flood. |
| AvLostLivesDueInjuries3months | The average number of lives lost due to injuries in the three months following the flood. |
| AverageLostLife | The average number of lives lost directly due to the flood. |
| AverageInjuries | The average number of injuries reported due to the flood. |

Source: FRMC.

The regression results for the 66 communities that experienced a flood while controlling for demographics and flood exposure and hazard, are presented next. Due to the small sample size, a significance level of 10% was considered relevant (Cameron and Trivedi, 2005; Deaton, 1997). We tested different specifications to analyse the sensitivity of the parameters to the inclusion of control variables (see Tables A10 to A15 in the appendices), but the preferred model for the analysis is the full model (with all controls). There is a vivid discussion among statisticians and econometricians on the role of control variables and whether they should be excluded if there is not statistical significance. This paper takes the stand that considering the control variables is relevant because they have theoretical meaning and, hence, even a non-significant result is a relevant result (Deaton, 2010; Pearl, 2000). Besides, as they help reduce omitted variable bias and improve causal inference (Angrist and Pischke, 2009). Overall, the full model displayed also better fit in all specifications (R-squared and AIC/BIC).

Another important issue for modelling is that, if the five resilience capitals or if the five DRM cycle stages are highly correlated, it can cause multicollinearity, making it difficult to determine their individual effects. High variance inflation factors (VIFs) would indicate if this is an issue. The results for the full model of the VIFs for the independent variables specified in the linear regression model shows that multicollinearity is present but not extreme, with mean VIF of 2.41. A mean VIF below 5 suggests that overall, a model is not suffering from severe multicollinearity (James et al., 2013).

**6.1 The Effect of Community Resilience on Health Outcomes**

Table 2 displays the results of the effect of the five resilience capitals on health outcomes (average deaths, average number of injuries, average number of deaths after three months). For first regression model, which analysed the effect of the five community resilience capitals on average deaths due to floods, no capital was found to be statistically significant. This lack of significant association is a critical finding. It suggests that certain forms of resilience, as currently measured, may not translate directly into reductions in flood-related mortality – or may only do so above a particular threshold of capital accumulation. Reporting such null results is essential to avoid publication bias and contributes to a more realistic understanding of the limits of resilience-building initiatives in extreme events. The only statistically significant variable at the 1% level was the control for the percentage of the community affected by the flood, in which a one-standard deviation increase in this variable was associated with an increase of 0.27 deaths, everything else held constant.



Next, Table 2 displays the results of the second regression model, which analysed the effect of the five community resilience capitals on average injuries to floods. Two main results emerged regarding our impact variables. Social capital was found to be strongly associated with a decline of injuries, with a one-standard deviation increase in this indicator reducing the average number of injuries in 39 units (statistically significant at 1% level). Also, Human capital was found to reduce injuries, with a one standard-deviation increase in this indicator leading to a drop of 5.97 injuries (statistically significant at 1% level). Regarding the controls, the average number of population aged 50 plus was also found to be negatively associated with injuries, with a one-standard deviation increase in this variable associated with a decrease in 9.7 injuries, *ceteris paribus* (statistically significant at 1% level). This might indicate that older individuals are more likely to evacuate early or take preventive measures before disasters, reducing their likelihood of flood-related injuries. Finally, the percentage of the community affected by the flood was found to increase the number of injuries, in which a one-standard deviation increase in this variable was associated with an increase of 18 injuries, everything else held constant.

Finally, Table 2 shows the results of the third regression model, which analysed the effect of the five community resilience capitals on average fatalities after three months of the flood event. This model revealed an unexpected result: natural capital scores were positively associated with delayed mortality, everything held constant, with a one standard-deviation increase in this variable associated with an increase in 1.59 delayed deaths (significant at 1% level). To further investigate this unexpected result, we ran a regression model with the same specification by cluster. The large coefficient for std_natural in Cluster 2 (9.387) could be driving the overall significant and positive effect. This makes theoretical sense, as natural capital in this cluster is low due to degraded natural environments and weak ecosystem services, even if some management efforts exist (Chapagain et al., 2024a). Because of this, small improvements may not help much, and the positive association with delayed mortality might reflect the overall vulnerability of these communities rather than a real benefit from natural capital.

Contrary to the model for injuries, the average number of population aged 50 plus was found to be positively associated with delayed mortality, with a one-standard deviation increase in this variable associated with an increase in 1.07 deaths after three months of a flood, *ceteris paribus* (statistically significant at 1% level). Older individuals may have a higher likelihood of developing complications from flood-related injuries, infections, or chronic disease exacerbation. Conditions such as cardiovascular disease, respiratory illnesses, and weakened immune function could make them more vulnerable to delayed mortality rather than immediate death.





Table 2: Wild Bootstrap Clustered Regression Models with the impact of 5Cs on health and mortality outcomes

| Independent variables | Dependent variables | | |
|---|---|---|---|
| | Average Fatalities | Average Injuries | Average Fatalities After Three Months |
| std_social | 0.054 | -39.976*** | -0.179 |
| std_financial | 0.203 | 15.602 | -3.341 |
| std_physical | -0.630 | 8.109 | 0.047 |
| std_human | -0.256 | -5.970*** | -0.500 |
| std_natural | 0.582 | -11.277 | 1.591*** |
| std_AverageAge15to25 | -0.721 | 1.510 | 0.285 |
| std_AverageAge50plus | -1.137 | -9.735*** | 1.079*** |
| std_AveragePercFemaleResp | 0.459 | -1.707 | -0.088 |
| std_AverageRural | 0.075 | 0.052 | 0.124 |
| std_AverageReturnPeriod | 0.685 | 27.150 | -0.645 |
| std_AveragePercComAffect | 0.271*** | 18.123*** | 1.797 |
| Observations | 66 | 66 | 66 |
| R-squared | 0.49 | 0.61 | 0.35 |
| AIC | 228.12 | 683.02 | 381.65 |
| BIC | 234.69 | 689.58 | 388.22 |

Observation: *** Significant at 1%; ** Significant at 5%; * Significant at 10%

Source: FRMC.

### 6.2 The Effect of DRM cycle stages on Health Outcomes

The results for the equations of the effects along the DRM cycle stages on average fatalities after a flood event
portray a different scenario from the resilience estimates presented before. We remember that we analyse
throughout the paper the results for Model 7 only, which contains all the control variables. Table 3 summarizes
the results. First, both Corrective Risk Reduction and Preparation scores were statistically associated with
average number of deaths. Surprisingly, Corrective Risk Reduction was positively associated with average
mortality, with a one-standard deviation increase in this score was associated with an increase in 0.65 deaths,
everything held constant. To further investigate this counterintuitive finding, we conducted a cluster-specific
regression analysis, which revealed substantial heterogeneity across community contexts. In Cluster 1, CRR
showed a small negative association (coefficient = -0.03, p = 0.925), while Cluster 2 exhibited a stronger
positive relationship (coefficient = 0.84, p = 0.396), suggesting this larger cluster may be driving the overall
significant effect. Cluster 2 has significant weaknesses across all measured DRM cycle dimensions. These
communities with low CRR scores might have implemented recent improvements that had not yet translated
into reduced mortality outcomes during our study period, potentially creating a lagged effect where reported
improvements coexist with historically high mortality rates.

Preparation scores were negatively associated with average mortality, with a one standard-deviation increase in
this indicator leading to a decrease in 0.53 deaths. This result is expected and aligns with disaster risk reduction
intuition and theories. Preparedness significantly reduces immediate flood mortality by enhancing early



response. Effective warning systems and safety knowledge help individuals take timely protective actions, minimizing exposure to life-threatening conditions. Well-developed emergency infrastructure and coordinated response efforts ensure that communities can react efficiently, preventing avoidable deaths. Additionally, strong community participation and external support improve rescue operations and medical aid delivery, further

reducing fatalities.

Finally, more control variables now display a significant relationship with the average number of deaths, being the average percentage of young population (15 to 25 years) and the average percentage of elderly (50 or more) negatively associated with the number of deaths (significant at 1% level). The explanations are quite straightforward: older individuals tend to take disasters more seriously, acting cautiously in response to early

warnings and evacuation orders. Life experience and risk awareness help them recognize the severity of floods and take protective measures earlier, reducing their chances of fatal exposure. Younger individuals tend to have better physical strength, endurance, and mobility, which increases their chances of escaping hazardous flood conditions. They are less likely to suffer from mobility limitations or pre-existing health conditions that make evacuation or survival difficult. Next, the average percentage of rural population was positively associated with

the number of deaths, with a one standard-deviation increase in this percentage related to 0.09 deaths. Factors such as socio-economic settings, water, sanitation condition, and state of public health infrastructure of rural areas can contribute to health complications and mortality (Jerin et al., 2024). Finally, flood hazard (return period) and exposure (percentage of community affected) are positively associated with mortality at the 1% of confidence level.

The model results for average injuries and its relationship with DRM cycle stages are also presented in Table 3. For this outcome variable, the results for Corrective Risk Reduction (CRR) and Preparedness are in line with the expectations: a one standard-deviation increase in CRR reduces the average number of injuries in 31 units; also, a one standard-deviation increase in Preparedness reduces the indicator in 23 units. Surprisingly, a one standard-deviation increase in the Prospective Risk Reduction (PRR) increases the number of injuries in 6 units. To

further investigate our counterintuitive finding regarding PRR, we again conducted a cluster-specific regression analysis, which revealed that in Cluster 1, which is characterized by high levels of overall preparedness, PRR maintained a significant positive relationship with injuries (coefficient = 4.88, p = 0.021), while in Cluster 2, the relationship was weaker and non-significant (coefficient = 2.93, p = 0.781). This pattern suggests that in communities with stronger overall disaster management systems (Cluster 1), there may be more effective injury

reporting and documentation mechanisms in place, leading to higher recorded injury rates despite better prevention measures. The results for control variables show that the average number of elderly is negatively associated with the number of injuries (significant at 1%), whereas flood hazard and exposure are positively associated with the number of injuries.

Finally, Table 3 presents the results for the impact along DRM cycle stages on delayed mortality from floods

(after 3 months). Results are in line and expected as per the literature regarding two DRM cycle stages: Prospective Risk Reduction (PRR) and Recovery. A one standard-deviation increase in the PRR score is associated with a decrease in 1.75 delayed deaths, everything held constant. Also, a one standard-deviation increase in the Recovery indicator is associated with a decline in 3.57 deaths, *ceteris paribus*. Surprisingly, the only significant control variables for delayed mortality are flood exposure (percentage of the community

affected) and hazard (return period), with a one-standard deviation increase in these indicators associated with a




decrease in delayed deaths. These results appear counterintuitive, as one would expect communities with more frequent disasters (shorter return periods) and greater affected populations to experience higher delayed mortality. Cluster-specific analysis revealed distinct patterns: in Cluster 1, neither variable showed significant relationships with delayed mortality (return period: coefficient = 1.98, p = 0.523; community affected: coefficient = 0.27, p = 0.877). However, in Cluster 2, both variables showed negative associations, with the community affected percentage approaching significance (coefficient = -1.98, p = 0.089) and return period showing a similar trend (coefficient = -1.42, p = 0.100). This may be associated with a survivorship bias or a "harvesting" effect: the most vulnerable individuals (e.g., the elderly, those with pre-existing health conditions) may succumb quickly after the flood, reducing the number of people who would die in the delayed mortality window. Alternatively, this finding may reflect a reporting phenomenon where communities with more frequent disasters have better systems for attributing later deaths to the original disaster event.

Table 3: Wild Bootstrap Clustered Regression Models with the impact of DRM cycle stages on health and mortality outcomes

| Independent variables | Dependent variables | | |
|---|---|---|---|
| | Average Fatalities | Average Injuries | Average Fatalities After Three Months |
| std_CRR | 0.656*** | -30.906*** | 1.285 |
| std_PREP | -0.532*** | -23.375*** | 0.637 |
| std_PRR | 0.022 | 5.921*** | -1.758*** |
| std_RECOV | -0.019 | 27.943 | -3.577*** |
| std_RESP | -0.419 | -3.674 | -0.031 |
| std_AverageAge15to25 | -0.744*** | 3.069 | -0.045 |
| std_AverageAge50plus | -0.925*** | -14.998*** | -0.030 |
| std_AveragePercFemaleResp | 0.490 | -3.284 | 0.137 |
| std_AverageRural | 0.094*** | -3.781 | 0.136 |
| std_AverageReturnPeriod | 0.845*** | 25.841*** | -0.458*** |
| std_AveragePercComAffect | 0.243*** | 13.702*** | -1.503*** |
| Observations | 66 | 66 | 66 |
| R-squared | 0.45 | 0.59 | 0.38 |
| AIC | 228.86 | 681.17 | 374.33 |
| BIC | 231.04 | 683.36 | 376.52 |

Observation: *** Significant at 1%; ** Significant at 5%; * Significant at 10%

Source: FRMC.

Interestingly, several DRM stages – including Corrective Risk Reduction and Preparedness – did not show statistically significant associations with delayed mortality. While these might initially appear as disappointing results, they offer crucial insights: either the health effects of floods evolve differently over time, or current DRM metrics may not fully capture interventions that affect medium-term health outcomes.



**7 Discussion and conclusions**

The goal of this paper was to explore the role of resilience and DRM cycle stages on health and mortality outcomes after flood events for 66 countries across seven countries participating in the Flood Resilience Measurement for Communities (FRMC). Namely, how latent capacities determine outcomes, in terms of reduced mortality and morbidity.

We advance current literature not only by incorporating a novel and rich dataset, but also by controlling in our impact analysis by relevant variables such as the demographic profile of the community and flood hazard and exposure. The literature stresses the role of these variables, and they are necessary to avoid confounding in the econometric analysis. Their significance in some models do endorse their relevance for the analysis, and the quality of the adjustment in the full models (with all controls) justify their inclusion.

Our results demonstrate the relevance of selected resilience scores and DRM cycle stages for health and mortality outcomes, with some unexpected results as well. Resilience models indicate that both social capital and natural capital are statistically, strongly, and negatively associated with the average number of injuries. However, apart from natural capital, none of the resilience indicators showed a statistically significant association with either immediate fatalities or fatalities occurring within three months, when controlling for all other factors. The absence of significant effects underscores the complexity of translating community capacity into lifesaving outcomes. These null results align with a growing body of development research that recognizes the importance of publishing and interpreting non-significant effects – not as failures, but as evidence that helps refine theories, interventions, and measurement tools. Surprisingly, natural capital scores were found to be strongly and positively associated with delayed mortality, an unexpected result that warrants further investigation. The control variable for flood exposure was statistically related to the number of deaths and injuries, while the average number of elderly individuals was negatively associated with the number of injuries but positively associated with delayed mortality.

In contrast, the DRM Cycle models demonstrate greater predictive power in terms of significant variables for health and mortality outcomes compared to resilience models, although some variables show unexpected effects. Preparedness emerged as the most relevant DRM stage, significantly leading to reductions in both immediate fatalities and injuries. The Corrective Risk Reduction stage was found to decrease injuries but unexpectedly increased fatalities. Recovery and Response were negatively associated with delayed mortality, aligning with existing literature on disaster risk management. Regarding control variables, both elderly and young populations were associated with a reduction in immediate fatalities, while the percentage of elderly individuals specifically contributed to a decrease in injuries. Also, a higher percentage of the rural population was positively correlated with the number of immediate deaths. For flood exposure and hazard indicators, these variables were strongly and positively associated with average fatalities and injuries but negatively associated with delayed mortality.

While prior research emphasizes their role in increasing health risks, this pattern might suggest that the most vulnerable individuals, such as older adults or those with preexisting health conditions, are more likely to succumb soon after the flood. Consequently, fewer individuals remain who could die later, leading to a lower number of delayed deaths.

The study has limitations that should be considered when interpreting the results. One key limitation is the small sample size, which may pose statistical challenges in estimating more complex models, such as zero-inflated specifications. Additionally, the study includes a small number of clusters, which can impact the reliability of



statistical inferences. To mitigate this issue, the analysis incorporates the use of wild bootstrap methods, providing a more robust approach to addressing the potential shortcomings in cluster representation.

Another limitation arises from the measurement of certain constructs, as they are not always well defined by a single indicator. This could introduce measurement errors, potentially affecting the accuracy and consistency of the results.


Finally, the demographic profile used in the study serves as a proxy variable for population characteristics, but it is representative only of household heads. This limitation may reduce the accuracy of the demographic analysis, as it does not fully capture the diversity and distribution of characteristics across the broader population.

Future research should expand the sample size to improve statistical power and allow for more complex modeling approaches. It should also explore the temporal dynamics of resilience and DRM interventions, particularly the lag between implementation and their impact on health outcomes. Additionally, improving demographic data availability would strengthen causal inference and help explain counterintuitive results.



**Appendices**

Table A1: Distribution of the communities affected by floods by country. FRMC data

| Country | Frequency |
| --- | --- |
| Bangladesh | 32 |
| El Salvador | 2 |
| Malawi | 8 |
| Mexico | 3 |
| Nepal | 5 |
| Senegal | 4 |
| Vietnam | 12 |
| Total | 66 |

Source: FRMC.

Table A2: Variables according to FRMC five capitals

| Financial | Human | Natural | Physical | Social capital |
| --- | --- | --- | --- | --- |
| Household asset recovery | Evacuation and safety knowledge | Natural capital condition | Flood healthcare access | Community participation in flood-related activities |
| Community disaster fund | First aid knowledge | Priority natural units | Early Warning Systems (EWS) | External flood response and recovery services |
| Business continuity | Education commitment during floods | Priority managed units | Flood emergency infrastructure | Community safety |
| Household income continuity strategy | Flood exposure awareness | Natural resource conservation | Provision of education | Community disaster risk management planning |
| Risk reduction investments | Asset protection knowledge | Natural habitat restoration | Household flood protection | Community structures for mutual assistance |
| Disaster response budget | Future flood risk awareness | | Large scale flood protection | Community representative bodies |
| Conservation budget | Water and sanitation awareness | | Transportation interruption | Social inclusiveness |
| | Environmental management awareness | | Communication interruption | Local leadership |
| | Governance awareness | | Flood emergency food supply | Inter-community flood coordination |
| | | | Flood safe water | Integrated flood management planning |
| | | | Flood waste contamination | National forecasting policy & plan |
| | | | Flood energy supply | |

Source: FRMC.

Table A3: Variables according to FRMC DRM Cycle

| Corrective Risk Reduction | Preparedness | Prospective Risk Reduction | Recovery | Response |
| --- | --- | --- | --- | --- |
| Risk reduction investments | Business continuity | Conservation budget | Household asset recovery | Disaster response budget |
| Asset protection | Evacuation and safety | Future flood risk | Community | Water and sanitation |



| | | | | |
|---|---|---|---|---|
| knowledge | knowledge | awareness | disaster fund | awareness |
| Governance awareness | First aid knowledge | Environmental management awareness | Provision of education | Flood healthcare access |
| Natural habitat restoration | Early Warning Systems (EWS) | Natural capital condition | Flood energy supply | Transportation interruption |
| Household flood protection | Flood emergency infrastructure | Priority natural units | Community safety | Communication interruption |
| Large scale flood protection | Community participation in flood-related activities | Natural resource conservation | | Flood emergency food supply |
| Community representative bodies | External flood response and recovery services | Community disaster risk management planning | | Flood safe water |
| Social inclusiveness | Inter-community flood coordination | Local leadership | | Flood waste contamination |
| Integrated flood management planning | National forecasting policy & plan | | | Community structures for mutual assistance |

Source: FRMC.

Table A4: Outcome variables for analysis

| Outcome variable | Description | Dataset | Respondents |
|---|---|---|---|
| Injuries | How many men in the community suffered serious injuries in the flood? How many women in the community suffered serious injuries in the flood? How many children in the community suffered serious injuries in the flood? How many men in the community suffered serious injuries in the flood? | PE | Key informant, focus group, secondary source |
| Deaths | How many men in the community lost their lives in the flood? How many women in the community lost their lives in the flood? How many children in the community lost their lives in the flood? | PE | Key informant, focus group, secondary source |
| Deaths after 3 months | Compared to the number of people who lose their lives from these illnesses in non-flood times, how many additional people lost their lives due to these illnesses in the 3 months following the flood? | PE | Key informant, focus group, secondary source |

Source: FRMC. Note: PE: Post-event survey

560                  Table A5: Control variables

| Control variable | Description | Dataset | Respondents |
|---|---|---|---|
| Average Percentage of Population Affected by Flood | What percentage of the community was directly impacted by the flood? | PE | Key informant, focus group, secondary source |
| Average Flood Return Period | What is the return period or re-occurrence interval of this flood, in number of years? In other words, how often is a flood of this size or bigger expected/experienced in the community? | PE | Key informant, focus group, secondary source |
| Age Group Distribution (%) | Which of the following age groups do you fall into: 15-25, 26-50, or over 50? | BL | Household |
| Gender Distribution (%) | What is your gender: Male, Female, Other? | BL | Household |





| Average Rural Composition (%) | Is this a rural, urban, or peri-urban community? | BL | Household |
|---|---|---|---|

Source: FRMC. Note: PE: Post-event survey; BL: Baseline survey

Table A6: Cronbach's alpha for the five capitals in the FRMC

| Capital | Number of items on the scale | Cronbach's alpha |
|---|---|---|
| Financial | 7 | 0.7710 |
| Social | 11 | 0.8453 |
| Physical | 12 | 0.8275 |
| Human | 9 | 0.7053 |
| Natural | 5 | 0.7022 |

Source: FRMC.


Table A7: Cronbach's alpha for the DRM cycle stages

| DRM Indicator | Number of items on the scale | Cronbach's alpha |
|---|---|---|
| Corrective Risk Reduction | 9 | 0.7365 |
| Preparedness | 9 | 0.7404 |
| Prospective Risk Reduction | 8 | 0.5646 |
| Recovery | 5 | 0.5830 |
| Response | 9 | 0.7727 |

Source: FRMC.







**Figure A1: Histograms for the FRMC five capitals**







Figure A2: Histograms for the DRM cycle stages





(a) Financial

(b) Social

(c) Physical

(d) Human

(e) Natural

**Figure A3: Screeplots for the FRMC five capitals**



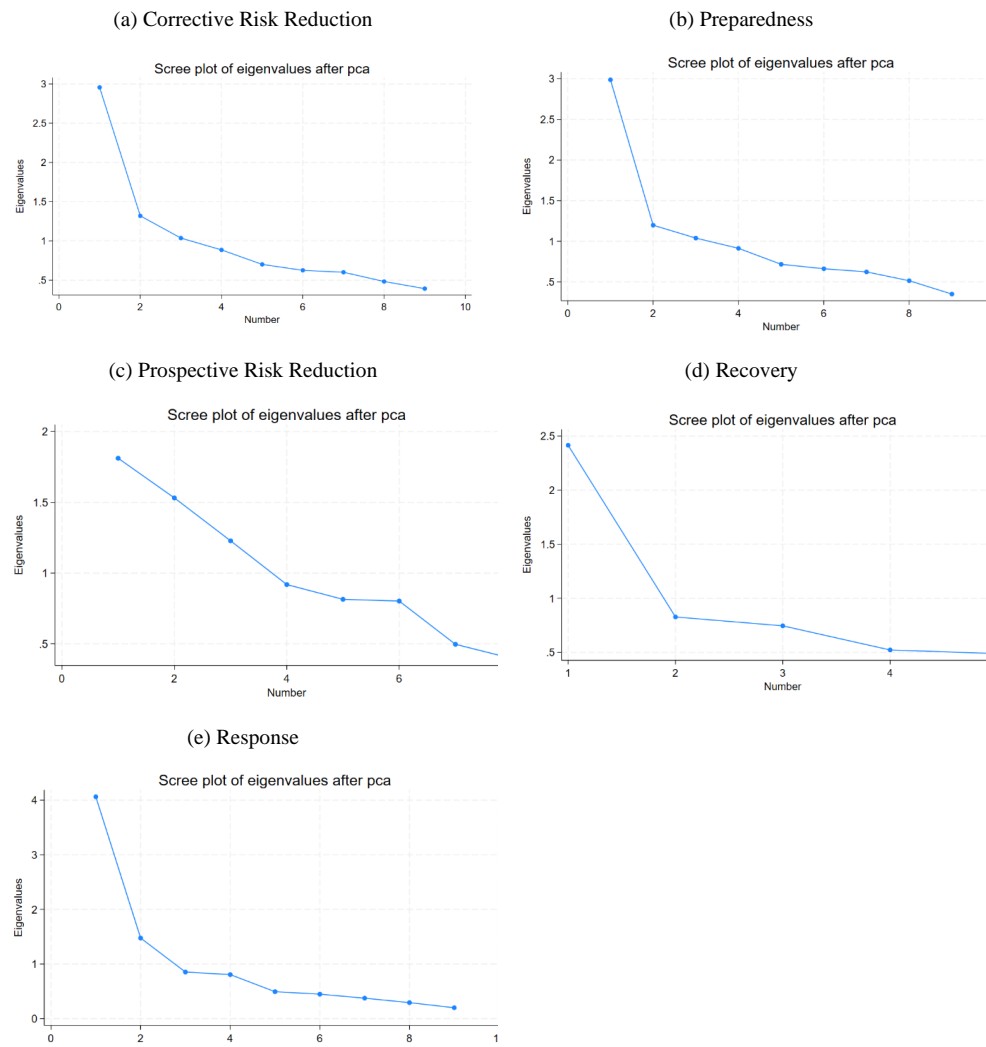

**Figure A4: Screeplot for the DRM cycle stages**

Table A8: Distribution of the 66 communities that have experienced flood according to resilience clusters

| Cluster | Freq. | Perc. | Cum. |
|---|---|---|---|
| 1 | 43 | 65.1 | 65.1 |
| 2 | 5 | 7.6 | 72.7 |
| 3 | 4 | 6.0 | 78.8 |
| 4 | 14 | 21.2 | 100.00 |
| Total | 66 | 100 | |

Source: FRMC


Table X: Distribution of the 66 communities that have experienced flood according to DRM cycle clusters

| Cluster | Freq. | Perc. | Cum. |
|---|---|---|---|



| | | | |
|---|---|---|---|
| 1 | 27 | 40.9 | 40.9 |
| 2 | 39 | 59.1 | 100.00 |
| Total | 66 | 100 | |

Source: FRMC

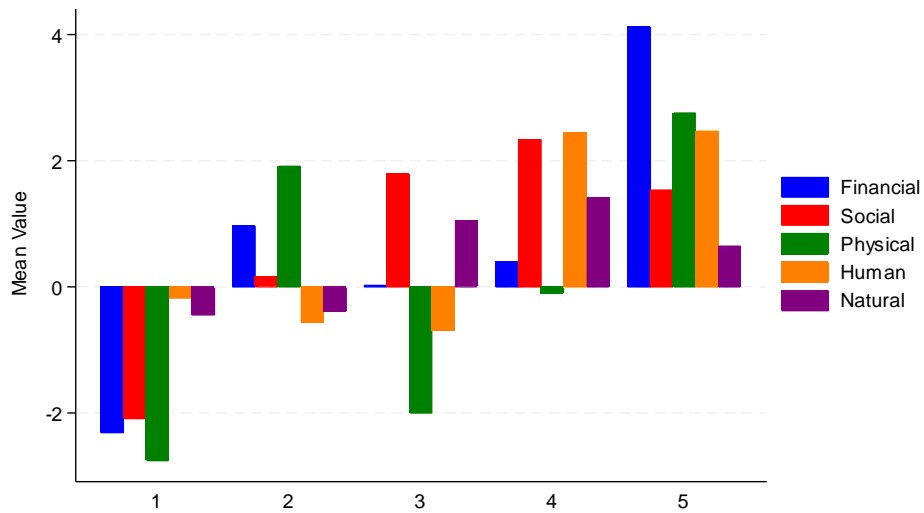


**Figure A5: Mean of resilience capitals scores by cluster in the baseline survey**

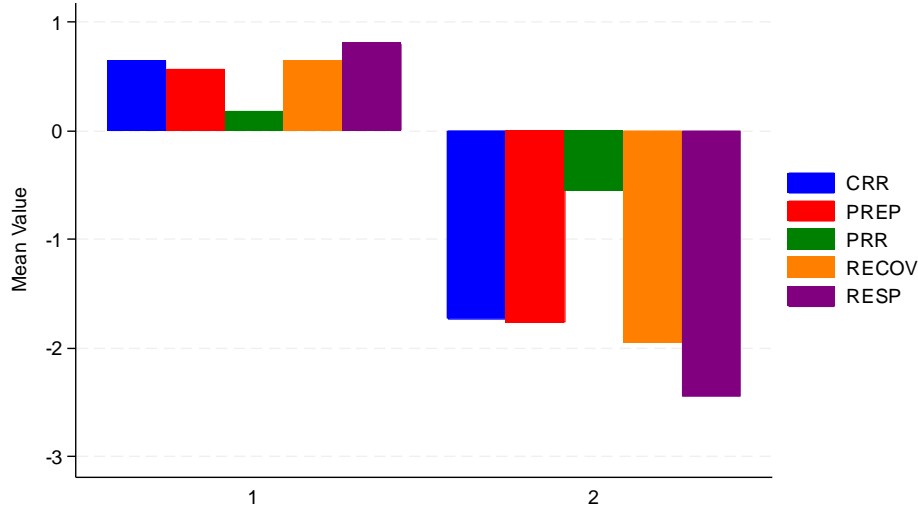

**Figure A6: Mean of DRM cycle stages scores by cluster in the baseline survey**




Table A9: Descriptive statistics for the control and dependent variables

| Variable | Observations | Mean | Std. Dev. | Min | Max |
|---|---|---|---|---|---|
| AverageAge15to25 | 66 | 0.100 | 0.079 | 0.000 | 0.324 |
| AverageAge26to50 | 66 | 0.586 | 0.185 | 0.184 | 0.928 |
| AverageAge50plus | 66 | 0.315 | 0.203 | 0.036 | 0.816 |
| AveragePercFemaleResp | 66 | 0.577 | 0.143 | 0.198 | 0.836 |
| AverageRural | 66 | 0.803 | 0.401 | 0.000 | 1.000 |
| AverageReturnPeriod | 66 | 8.111 | 8.793 | 1.000 | 35.000 |
| AveragePercComAffect | 66 | 0.710 | 0.225 | 0.183 | 1.000 |
| AvLostLivesDueInjuries3months | 66 | 1.695 | 4.458 | 0.000 | 23.000 |
| AverageLostLife | 66 | 1.045 | 1.978 | 0.000 | 11.000 |
| AverageInjuries | 66 | 48.668 | 68.475 | 0.000 | 257.750 |

Source: FRMC.



Table A10: Comparison of Wild Bootstrap Clustered Regression Models for Assessing Community Resilience and Its Impact on Average Loss of Life

| Variable | Model 1 | Model 2 | Model 3 | Model 4 | Model 5 | Model 6 | Model 7 |
|---|---|---|---|---|---|---|---|
| std_social | -0.537*** | -0.371 | 0.190 | 0.191 | 0.097 | 0.112 | 0.054 |
| std_financial | 0.260 | 0.053 | 0.215 | 0.322 | 0.391 | 0.218 | 0.203 |
| std_physical | -0.583 | -0.706 | -0.846 | -0.917 | -0.854 | -0.755 | -0.630 |
| std_human | -0.472 | -0.609 | -0.601 | -0.466 | -0.468 | -0.229 | -0.256 |
| std_natural | 0.434 | 0.464 | 0.644 | 0.618 | 0.659 | 0.543 | 0.582 |
| std_AverageAge15to25 | | -0.449 | -0.529 | -0.650 | -0.651 | -0.744 | -0.721 |
| std_AverageAge50plus | | | -0.887* | -1.011 | -0.961 | -1.146 | -1.137 |
| std_AveragePercFemaleResp | | | | 0.409 | 0.464 | 0.432 | 0.459 |
| std_AverageRural | | | | | 0.236 | 0.131 | 0.075 |
| std_AverageReturnPeriod | | | | | | 0.695 | 0.685 |
| std_AveragePercComAffect | | | | | | | 0.271*** |
| Observations | 66 | 66 | 66 | 66 | 66 | 66 | 66 |
| R-squared | 0.23 | 0.27 | 0.35 | 0.38 | 0.39 | 0.48 | 0.49 |
| AIC | 255.23 | 251.66 | 243.44 | 240.39 | 239.63 | 229.43 | 228.12 |
| BIC | 261.79 | 258.23 | 250.01 | 246.96 | 246.20 | 236.00 | 234.69 |

Observation: *** Significant at 1%; ** Significant at 5%; * Significant at 10%

Source: FRMC.





Table A11: Comparison of Wild Bootstrap Clustered Regression Models for Assessing Community Resilience and Its Impact on Average Injuries

| Variable | Model 1 | Model 2 | Model 3 | Model 4 | Model 5 | Model 6 | Model 7 |
|---|---|---|---|---|---|---|---|
| std_social | -36.145 | -37.254 | -33.537 | -33.547 | -36.736 | -36.127*** | -39.976*** |
| std_financial | 19.833 | 21.218 | 22.288 | 21.208 | 23.513 | 16.631 | 15.602 |
| std_physical | -6.930 | -6.104 | -7.032 | -6.320 | -4.203 | -0.243 | 8.109 |
| std_human | -13.281 | -12.361 | -12.310 | -13.667 | -13.742 | -4.156 | -5.970*** |
| std_natural | -11.879*** | -12.076*** | -10.884*** | -10.627*** | -9.240*** | -13.873 | -11.277 |
| std_AverageAge15to25 | | 3.013 | 2.486 | 3.704 | 3.672 | -0.068 | 1.510 |
| std_AverageAge50plus | | | -5.878 | -4.633 | -2.935 | -10.352 | -9.735*** |
| std_AveragePercFemaleResp | | | | -4.106*** | -2.251 | -3.516 | -1.707 |
| std_AverageRural | | | | | 7.992*** | 3.786 | 0.052 |
| std_AverageReturnPeriod | | | | | | 27.799 | 27.150 |
| std_AveragePercComAffect | | | | | | | 18.123*** |
| Observations | 66 | 66 | 66 | 66 | 66 | 66 | 66 |
| R-squared | 0.45 | 0.45 | 0.46 | 0.46 | 0.46 | 0.57 | 0.61 |
| AIC | 705.24 | 705.07 | 704.72 | 704.44 | 703.68 | 688.74 | 683.02 |
| BIC | 711.81 | 711.64 | 711.29 | 711.01 | 710.25 | 695.31 | 689.58 |

Observation: *** Significant at 1%; ** Significant at 5%; * Significant at 10%
Source: FRMC.



Table A12: Comparison of Wild Bootstrap Clustered Regression Models for Assessing Community Resilience and Its Impact on Average Fatalities After Three Months

| Variable | Model 1 | Model 2 | Model 3 | Model 4 | Model 5 | Model 6 | Model 7 |
|---|---|---|---|---|---|---|---|
| std_social | 0.096 | -0.011 | -0.687 | -0.687 | -0.546 | -0.561 | -0.179 |
| std_financial | -3.493 | -3.359 | -3.554 | -3.517 | -3.619 | -3.443 | -3.341 |
| std_physical | 0.846*** | 0.925*** | 1.094*** | 1.070 | 0.976 | 0.875 | 0.047 |
| std_human | -0.565 | -0.476 | -0.485 | -0.439 | -0.436 | -0.680 | -0.500 |
| std_natural | 2.037*** | 2.017*** | 1.801 | 1.792 | 1.730 | 1.849 | 1.591*** |
| std_AverageAge15to25 | | 0.291 | 0.387 | 0.345 | 0.346 | 0.442 | 0.285 |
| std_AverageAge50plus | | | 1.069*** | 1.026*** | .951*** | 1.140*** | 1.079*** |
| std_AveragePercFemaleResp | | | | 0.141 | 0.059 | 0.091 | -0.088 |
| std_AverageRural | | | | | -0.354 | -0.246 | 0.124 |
| std_AverageReturnPeriod | | | | | | -0.710 | -0.645 |
| std_AveragePercComAffect | | | | | | | 1.797 |
| Observations | 66 | 66 | 66 | 66 | 66 | 66 | 66 |
| R-squared | 0.27 | 0.27 | 0.28 | 0.28 | 0.29 | 0.30 | 0.35 |
| AIC | 389.93 | 389.74 | 388.34 | 388.30 | 388.12 | 387.08 | 381.65 |
| BIC | 396.50 | 396.31 | 394.91 | 394.87 | 394.69 | 393.65 | 388.22 |

Observation: *** Significant at 1%; ** Significant at 5%; * Significant at 10%
Source: FRMC.



Table A13: Comparison of Wild Bootstrap Clustered Regression Models for Assessing DRM cycle levels and Its Impact on
Average Fatalities

| Variable | Model 1 | Model 2 | Model 3 | Model 4 | Model 5 | Model 6 | Model 7 |
|---|---|---|---|---|---|---|---|
| std_CRR | -.015*** | 0.100 | 0.315*** | 0.335*** | 0.613*** | 0.716*** | 0.656*** |
| std_PREP | -.577*** | -0.550*** | -0.460*** | -0.491*** | -0.699*** | -0.597*** | -0.532*** |
| std_PRR | 0.095 | 0.131 | 0.022 | 0.025 | 0.029 | 0.068 | 0.023 |
| std_RECOV | 0.482 | 0.062 | 0.299*** | 0.314*** | 0.874*** | 0.079*** | -0.019 |
| std_RESP | -.774*** | -0.640*** | -0.743*** | -0.734*** | -1.063*** | -0.563*** | -0.420 |
| std_AverageAge15to25 | | -0.344*** | -0.312*** | -0.313*** | -0.447*** | -0.741*** | -0.744*** |
| std_AverageAge50plus | | | -0.528*** | -0.516*** | -0.733*** | -0.901*** | -0.925*** |
| std_AveragePercFemaleResp | | | | 0.599 | 0.676 | 0.491 | 0.491 |
| std_AverageRural | | | | | 0.278*** | 0.172*** | 0.094*** |
| std_AverageReturnPeriod | | | | | | 0.832*** | 0.845*** |
| std_AveragePercComAffect | | | | | | | 0.243*** |
| Observations | 66 | 66 | 66 | 66 | 66 | 66 | 66 |
| R-squared | 0.20 | 0.22 | 0.25 | 0.31 | 0.32 | 0.44 | 0.45 |
| AIC | 253.52 | 251.91 | 249.00 | 243.61 | 242.67 | 229.72 | 228.86 |
| BIC | 255.71 | 254.10 | 251.19 | 245.80 | 244.86 | 231.91 | 231.04 |

Observation: *** Significant at 1%; ** Significant at 5%; * Significant at 10%
Source: FRMC.






Table A14: Comparison of Wild Bootstrap Clustered Regression Models for Assessing DRM cycle levels and Its Impact on Average Injuries

| Variable | Model 1 | Model 2 | Model 3 | Model 4 | Model 5 | Model 6 | Model 7 |
|---|---|---|---|---|---|---|---|
| std_CRR | -32.123*** | -36.119*** | -32.668*** | -32.294*** | -30.654*** | -27.559 | -30.906*** |
| std_PREP | -28.290*** | -29.227*** | -27.792*** | -27.963*** | -30.126*** | -27.041*** | -23.375*** |
| std_PRR | 10.098*** | 8.831*** | 7.077*** | 7.057*** | 7.290*** | 8.478*** | 5.921*** |
| std_RECOV | 36.256*** | 50.975 | 54.779 | 55.717 | 57.476 | 33.488 | 27.944 |
| std_RESP | -19.805*** | -24.528*** | -26.193*** | -26.870*** | -26.831*** | -11.777 | -3.675 |
| std_AverageAge15to25 | | 12.051 | 12.555 | 12.306 | 12.081 | 3.224 | 3.070 |
| std_AverageAge50plus | | | -8.446*** | -8.934*** | -8.557*** | -13.617*** | -14.998*** |
| std_AveragePercFemaleResp | | | | 1.259 | 2.309 | -3.256*** | -3.285 |
| std_AverageRural | | | | | 3.814*** | 0.593 | -3.782 |
| std_AverageReturnPeriod | | | | | | 25.092*** | 25.841*** |
| std_AveragePercComAffect | | | | | | | 13.702*** |
| Observations | 66 | 66 | 66 | 66 | 66 | 66 | 66 |
| R-squared | 0.46 | 0.48 | 0.49 | 0.49 | 0.49 | 0.58 | 0.59 |
| AIC | 699.54 | 697.23 | 696.38 | 696.35 | 696.17 | 684.03 | 681.17 |
| BIC | 701.73 | 699.42 | 698.57 | 698.54 | 698.36 | 686.22 | 683.36 |

Observation: *** Significant at 1%; ** Significant at 5%; * Significant at 10%

Source: FRMC.





Table A15: Comparison of Wild Bootstrap Clustered Regression Models for Assessing DRM cycle levels and Its Impact on Average Fatalities After Three Months

| Variable | Model 1 | Model 2 | Model 3 | Model 4 | Model 5 | Model 6 | Model 7 |
|---|---|---|---|---|---|---|---|
| std_CRR | 0.956 | 1.020*** | 1.085 | 1.132 | 0.964 | 0.917 | 1.285 |
| std_PREP | 0.843 | 0.858 | 0.885 | 0.863 | 1.085*** | 1.039*** | 0.637 |
| std_PRR | -1.983*** | -1.962*** | -1.995*** | -1.997*** | -2.021*** | -2.039*** | -1.758*** |
| std_RECOV | -4.314*** | -4.553*** | -4.482*** | -4.365*** | -4.545*** | -4.185*** | -3.577*** |
| std_RESP | 1.127*** | 1.204*** | 1.173*** | 1.088*** | 1.084*** | 0.858*** | -0.031 |
| std_AverageAge15to25 | | -0.196 | -0.187 | -0.218 | -0.195 | -0.062 | -0.045 |
| std_AverageAge50plus | | | -0.157 | -0.219 | -0.257 | -0.181 | -0.030 |
| std_AveragePercFemaleResp | | | | 0.158 | 0.050 | 0.134 | 0.137 |
| std_AverageRural | | | | | -0.392 | -0.344 | 0.136 |
| std_AverageReturnPeriod | | | | | | -0.376 | -0.458*** |
| std_AveragePercComAffect | | | | | | | -1.503*** |
| Observations | 66 | 66 | 66 | 66 | 66 | 66 | 66 |
| R-squared | 0.34 | 0.34 | 0.34 | 0.34 | 0.35 | 0.35 | 0.38 |
| AIC | 378.63 | 378.55 | 378.51 | 378.46 | 378.22 | 377.91 | 374.33 |
| BIC | 380.82 | 380.74 | 380.70 | 380.65 | 380.41 | 380.10 | 376.52 |

Observation: *** Significant at 1%; ** Significant at 5%; * Significant at 10%

Source: FRMC.



**Code availability**

Code available upon request.

**Data availability**

The FRMC data is owned by the organisations that collected it. The authors of this paper were granted
access to the data but do not have permission to share it further.

**Author contribution**

RG designed the study, wrote the manuscript, and prepared and analysed the data. RM contributed to the
study design, manuscript writing, and data interpretation. SV participated in data preparation, manuscript
writing, and data interpretation. DC contributed to the writing of the manuscript and to data interpretation.

**Competing interests**

The authors declare the following financial interests/personal relationships which may be considered as
potential competing interests: RG reports financial support was provided by Z Zurich Foundation.

**Acknowledgments**

The authors thank the Z Zurich Foundation, Switzerland for providing funding.

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
