# Peer review of "The Effect of Community Resilience and Disaster Risk Management Cycle Stages on Morbi-Mortality Following Floods: An Empirical Assessment"

_EGUsphere, 2025_

## Author Response (AR1)

**Dear Reviewers and Editor,**

We sincerely thank you for your thoughtful and constructive comments, which have greatly helped improve the quality of our paper.

In this document, we provide detailed responses to each of the comments, along with a version of the manuscript with track changes for your review.

**Response to Reviewer #1**

Dear Anonymous Referee #1,

Thank you for taking the time to review our manuscript and provide feedback. We respectfully address your comments as follows:

1. *"[The paper] does not present a systematic methodology to address the research question and relies solely on basic statistical analysis based on the collected data."*
   We respectfully disagree with this assessment. Our study presents a clear and systematic methodological strategy to explore the relationship between health outcomes and key explanatory variables – namely, community resilience indicators and Disaster Risk Management (DRM) cycle stages – while controlling for potential confounders. This approach is consistent with established empirical practices aimed at reducing omitted variable bias in observational studies.
   Moreover, we contest the characterization of our statistical analysis as "basic." While our primary estimation strategy is based on Ordinary Least Squares (OLS), this choice was made deliberately, as it is well-suited to the data structure and research question. Also, our OLS approach is comprehensive in a targeted manner given the research question. Our methodology includes construct validation, descriptive analysis, and multivariate regression modeling. We also explored more advanced estimators; however, due to sample size constraints, those models yielded unstable results, which we clearly acknowledge in the manuscript. We maintain that the strategy adopted is both statistically sound and appropriate for the empirical setting.

2. *"Given the substantial differences among the seven countries—across social, economic, and infrastructural dimensions—the use of such simplified statistical methods lacks the robustness required to account for these complex factors."*
   We acknowledge that cross-country heterogeneity poses an important analytical challenge. To mitigate this, we adopted several strategies. In addition to including controls for demographic and hazard/exposure-related factors, we adjusted standard errors using the classification developed by Chapagain et al. (2024), which captures patterns in resilience across different community profiles. While it is not possible to completely eliminate unobserved heterogeneity, these steps represent a careful and methodologically sound effort to account for underlying structural differences. Our

study also relies on the Flood Resilience Measurement for Communities (FRMC), a well-established framework designed to generate empirical insights into resilience in diverse settings (Keating et al. 2017; Hochrainer-Stigler et al. 2020; Laurien et al. 2020; Keating et al. 2025). The FRMC uses a consistent set of indicators across all study sites, ensuring comparability, but allows flexibility in how data are gathered to reflect local realities. Data collection is carried out in partnership with local stakeholders, ensuring that the perspectives and experiences of communities are embedded in both the process and outcomes (Hochrainer-Stigler et al. 2021). This combination of standardization and adaptability supports meaningful cross-site comparisons. Chapagain et al. further demonstrated that the FRMC successfully identifies distinct community types through statistical clustering, confirming its ability to capture variations in flood resilience across different contexts.

In light of these considerations, we kindly request that the editor take this clarification into account when assessing the manuscript for further consideration.

Thank you. We had extensively revised the reference list for missing information.

**Response to Editor**

Dear Editor,

We thank you for considering our article for publication and for the comprehensive comments, which are addressed below.

*Dear Authors,*

*I have careful read your article as well as the reviews kindly provided by the two Reviewers.*

*Regarding the comment made by Reviewer 1, stating that "However, it does not present a systematic methodology to address the research question and relies solely on basic statistical analysis based on the collected data," I must say I agree.*

*I believe the structure of the paper makes it difficult to clearly identify the methodological steps, as they are scattered across various paragraphs and interspersed with numerous bibliographic references.*

*I suggest revising the overall structure to more clearly distinguish between the literature review, the methodology, and the results.*

We appreciate the editor's suggestions, which have significantly improved the overall flow of the paper. We have thoroughly revised the manuscript to improve its structure, consolidating the literature review and methodological steps, which were previously dispersed across the Methods and Results sections.

*For example, you might start by listing the key factors identified in the literature (preferably as bullet points), followed by a clear, reference-free description of the methodology employed in your study.*

We have substantially restructured and streamlined the literature review to improve the flow, using bullet points where appropriate (i.e., when listing multiple elements). In addition, references previously included in the methodology section have been removed.

*Moreover, a flowchart illustrating the methodology would make it easier to follow the various steps and would certainly do justice to such an extensive and valuable piece of work.*

Thank you for the suggestion. We have now included a flowchart illustrating the three main methodological steps of the study.

*I would also encourage the authors not to relegate all the material needed to understand the methodology to the appendix. At least the initial tables—those that explain the study places parameters used—should be included in the main text.*

This suggestion was well received and made good sense. Although the number of tables in the main text has increased considerably, their inclusion now provides a clearer and more logical structure for the reader to follow.

*Ideally, they should be careful reformatted for clarity and conciseness, maybe put in landscape format, avoiding redundancies (for instance, the phrase "in the community" appears several times unnecessarily). Similarly, I suggest revising the diagrams, as it is not always clear what the axes represent.*

As landscape-format tables are not permitted by the journal's style guidelines, we have adjusted all tables accordingly. Additionally, we revised the text to eliminate redundancies and ensured that all diagrams include properly labeled axes.

*In the discussion/conclusions as well, I would encourage the authors to make an effort to better finalize and synthesize the text—possibly by adding a diagram or figure that offers a clear, overarching interpretation of the work.*

We agree with the editor's suggestion and have included a table summarizing the results. The table highlights statistically significant effects. We also extensively revised the text in the section.

*As it stands, much of the effort seems to be relegated to the appendix, without sufficient attention given to facilitating its understanding and practical use.*

We believe that now the appendix reflects only the supplementary materials for those interested in supporting information.

*As minor comments,*

*• I suggest that the authors define each acronym only upon its first appearance in the text, as the current version includes repeated definitions, which may interrupt the flow of reading.*

We appreciate this suggestion. All acronyms are now defined only once upon first mention and consistently used throughout the text thereafter.

*• In Section 2.1, the five capitals should be presented as bullet points for clarity, and the remainder of the paragraph should be streamlined and better structured, as it is currently somewhat confusing.*

Thank you for the suggestions. The five capitals and the DRM cycle stages are now presented as bullet points to improve clarity and readability.

*• In Section 3.2, I recommend adding a brief explanation of the concept of flood return period, as not all readers may be familiar with this technical term.*

We added the following definition "The flood return period refers to the time (in years) that elapsed between two events that equal or exceed a particular magnitude (Paul and Mahmood, 2016)."

*I would like to invite you to submit a revised version of your manuscript according to my suggestions and the suggestions of both referees.*

*Would you please also provide an 'author's reply' to the reviewers (feel free to use the same words that you used in what you have already uploaded). Please, also include a track changes document between the old manuscript and the new one (you can include this as part of your 'author's reply').*

We have provided the requested document as part of this submission.

*I look forward to seeing the next version of your manuscript which I will not send out for further review, but rather, will make the decision myself, assuming no major items come up in the revised manuscript for which I need outside reviewers to aid me in my decision.*

*Best regards*

[revised manuscript text omitted]

ity suffered

seriousinjuries

injuries

in

the

flooded

2. How

many

women

in

the community suffered serious injuries in the flooded.2 How ma

ny children in the community suffered serious injuries in th

e
flood
2How many men in the community suffered serious injur

...ies in the flooded 2

Deaths | How many men in the community lost their li... | PE | Key informant; focus group, secondary sources...

yes e

. in the flood?

2 How many women in the community lost their

lives in the flood2How many children in the community lost t

heir lives in the flood 2

Compared to the number of people who

Key informment, focus group, secondar

[revised manuscript text omitted]